# Species-Specific Secondary Metabolites from *Primula veris* subsp. *veris* Obtained In Vitro Adventitious Root Cultures: An Alternative for Sustainable Production

Virginia Sarropoulou [1], Eirini Sarrou [1], Andrea Angeli [2], Stefan Martens [2], Eleni Maloupa [1] and Katerina Grigoriadou [1,*]

1 Balkan Botanic Garden of Kroussia-Laboratory for the Conservation and Evaluation of Native and Floricultural Species, Institute of Plant Breeding and Genetic Resources, Hellenic Agricultural Organization-DIMITRA, Thermi, P.O. Box 60458, GR-570 01 Thessaloniki, Greece
2 Fondazione Edmund Mach, Center of Research and Innovation, San Michele all' Adige, 38098 Trento, Italy
* Correspondence: katgrigoriadou@elgo.gr; Tel.: +30-9-2310-471110

**Abstract:** *Primula veris* subsp. *veris* L. is a perennial herbaceous and medicinal plant species the roots and flowers of which are a source of valuable pharmaceutical raw materials. The plant tissues are used to produce expectorant and diuretic drugs due to their high content of triterpene saponins and phenolic glycosides. Underground roots of *P. veris* can be obtained only through a destructive process during the plant's harvesting. In the present study, an in vitro adventitious root production protocol was developed as an alternative way of production, focused on four species-specific secondary metabolites. Root explants were cultured in Murashing & Skoog liquid medium supplemented with 5.4 μM α-naphthaleneacetic acid, 0.5 μM kinetin, L-proline 100 mg/L, and 30 g/L sucrose, in the dark and under agitation. The effect of temperature (10, 15 and 22 °C) on biomass production was investigated. The content of two flavonoid compounds (primeverin and primulaverin), and two main triterpene saponins (primulic acid I and II) were determined after 60 days of culture and compared with 1.5-year-old soil-grown plants. The accumulated content (mg/g DW) of bioactive compounds of in vitro adventitious roots cultured under 22 °C was significantly higher than the other two temperatures of the study, being 9.71 mg/g DW in primulaverin, 0.09 mg/g DW in primeverin, 6.09 mg/g DW in primulic acid I, and 0.51 mg/g DW in primulic acid II. Compared to the soil-grown roots (10.23 mg/g DW primulaverin, 0.28 mg/g DW primeverin, 17.01 mg/g DW primulic acid I, 0.09 mg/g DW primulic acid II), the in vitro grown roots at 22 °C exhibited a 5.67-fold higher content in primulic acid II. However, primulic acid I and primeverin content were approximately three-fold higher in soil-grown roots, while primulaverin content were at similar levels for both in vitro at 22 °C and soil-grown roots. From our results, tissue culture of *P. veris* subsp. *veris* could serve not only for propagation but also for production of species-specific secondary metabolites such as primulic acid II through adventitious root cultures. This would therefore limit the uncontrolled collection of this plant from its natural environment and provide natural products free from pesticides in a sustainable way.

**Keywords:** methylated flavonoid glycosides; primeverin; primulaverin; primulic acids; triterpene saponins; UPLC-MS/MS; MRM analyses

## 1. Introduction

*Primula veris* subsp. *veris* L. (syn. *Primula officinalis* Hill, *Primulaceae*) with common name cowslip is a perennial herbaceous and medicinal plant species (Figure 1a,b). The roots and flowers, which are a source of valuable pharmaceutical raw materials, are used to produce expectorant and diuretic drugs because of their high content in triterpene saponins and phenolic glycosides [1,2]. Extracts from cowslip roots are components of many herbal preparations such as Bronchicum, Pectosol, Tussipect, and Sinupret [3,4]. The safety and

efficacy of cowslip extracts rich in saponins have been demonstrated in several pharmacological studies, which show their potent anti-asthmatic, anti-inflammatory and anti-viral properties [5]. According to the exiting EMA monograph (EMA/HMPC/104095/2012) [6], primula radix (root) preparations can be used as an expectorant for coughs associated with colds, and the main indication for primula root is for the treatment of respiratory tract problems, such as asthma, bronchitis, and catarrh; these actions are attributed to secretolytic and secretomotoric triterpenoid saponins, which are present in the plant material in amounts up to 12% [7].

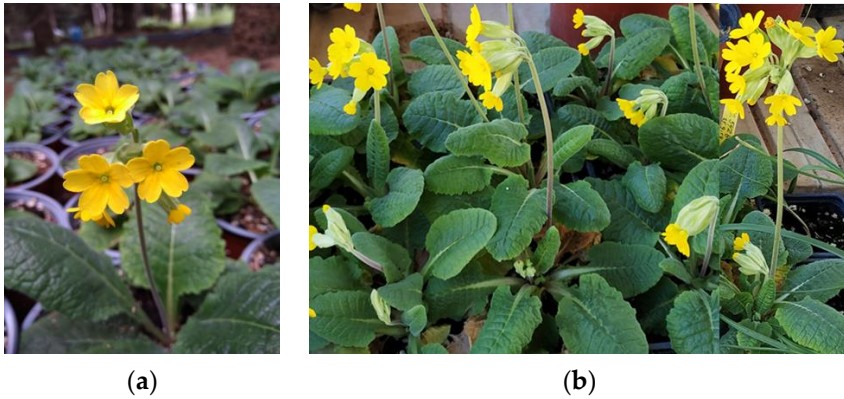

(**a**)     (**b**)

**Figure 1.** (**a**) *Primula veris* subsp. *veris* plant material: (**a**) In vivo pot (2.5 L) plants used for the evaluation of their root biomass production and their content in secondary metabolites; (**b**) Plants collected from a wild habitat, Northern Greece in Mt Pangaio, and transferred for maintenance at the Balkan Botanic Garden of Kroussia of the Institute of Plant Breeding and Genetic Recourses.

Apart from its role in human health, *P. veris* seems to be an important plant species for the natural habitats as it can serve as an early indicator of an ecosystem's health and quality by responding rapidly to direct negative environmental changes [8]. Therefore, the preservation of the species in wild habitats appears to be very important; however, the survival of the plant undergoes significant risks due to severe environmental changes together with overexploitation of the natural population by humans. For instance, in Greece, large amounts of plant biomass are illegally collected from the wild habitats early in spring during flowering period and sold to the trade market. This overexploitation of the species from nature has resulted in the shrinking and/or fragmentation of the natural populations and has directly affected environmental sustainability. Therefore, different methods for the sustainable production of *P. veris* plant biomass should be adopted for the exploitation of its valuable bioactive secondary metabolites.

Recent studies have demonstrated that the habitats and the respective abiotic factors influence the flavonoid glycoside pattern in the petals and leaves of *P. veris* [9]. However, only the petals and roots are used pharmaceutically, based on the expectorant and secretolytic effects of the triterpenoid saponins, such as priverosaponin B-22-acetate, primulic acid I and primulic acid II [10]. The valuable secondary metabolites (SMs) produced in the underground organs of *P. veris* can be obtained only from cultivated plants [11]. In the majority of *Primula* species, including *P. veris*, there are well-known difficulties with seed germination under greenhouse or field conditions [11,12], therefore, alternative ways such as in vitro culture have been suggested for producing plant raw material for the pharmaceutical industry from the protected plant species in a relatively short time [13].

Plant roots are one of the main raw materials used for herbal drug preparations, accounting for about 60% of herbal medicinal plants used in ethnomedicine [14]. Adventitious root cultures in vitro are highly useful due to their capability and/or potential for micropropagation and germplasm preservation [14]. At a commercial level, part of the production of important SMs is mainly performed in vitro through root cultures [15,16]. The roots of many medicinally important plants serve as an origin of various bioactive molecules that consist of diverse metabolites, proteins, agrochemicals, flavors, fragrances,

and dyes [17]. Furthermore, plant roots grown from inoculums and cultured in a plant-growth-regulator-amended medium are highly stable and synthesize ample amounts of plant metabolites in the intercellular spaces, which can easily be further extracted [18]. Some of the advantages of using plant cell, tissue, and organ cultures to produce root biomass in vitro are the higher growth rates, the shorter time period, and the low amount of inoculum in culture medium while stable production of the targeted metabolites [18]. Availability, overexploitation, difficulties in cultivation of the source plant, low productivity, phyto-geographical and seasonal variation in productivity, tissue and/or organ-specific production, difficulties in purification, variability of impurities, and the economic cost involved in the selection and implementation of appropriate screening bioassays are limiting factors for the industrial production of these phytochemicals from field-grown plants, and chemical synthesis is often not economically feasible because of the plants' highly complex structures and stereospecificity [18–20]. The adventitious root culture is considered an effective technique in producing constant biomass and secondary metabolites [21]. The tissue culture of some *Primula* species, as an alternative way to produce SMs, has been previously suggested [13], however the information for in vitro potential root biomass production is fragmented.

The aim of the present study was to investigate if an in vitro adventitious root production protocol in liquid media could serve as an alternative method of biomass production. For this purpose, our study was focused on four species-specific secondary metabolites in the group of triterpene saponins and flavonoids.

## 2. Materials and Methods

### 2.1. Plant Material and In Vitro Culture Conditions

Initial plant propagable material (plants and seeds) were collected from Northern Greece (Mt Pangaio, altitude of 1750 m a.s.l.) (Figure 1a). The collected plant material was maintained at Balkan Botanic Garden of Kroussia (BBGK) of the Institute of Plant Breeding and Genetic Recourses (Figure 1b) and received the IPEN (International Plant Exchange Network) accession number GR-1-BBGK-10,5428.

Stock shoot-tip in vitro cultures were established after the disinfection and germination (at a 50% rate) of 8-year old cold-storage (4 °C, RH < 5%) seeds after a culture period of 60 days following the protocol described by Grigoriadou et al. [22]. The culture medium used for the initial establishment and germination of seeds was the MS (Murashige and Skoog) [23] supplemented with 250 mg/L gibberellic acid ($GA_3$), 20 g/L sucrose, and 6 g/L Plant Agar (pH: 5.8) [17]. The derived shoot-tip cultures were afterwards sub-cultured every 4 weeks in MS medium enriched with 0.88 μM 6-benzyladenine (BA), 0.1 μM indole-3-butyric acid (IBA), 30 g/L sucrose, and 6 g/L Plant Agar (pH:5.8) for proliferation and rooting under $22 \pm 1$ °C, 16 h photoperiod, 40 μmol $m^{-2} s^{-1}$, and cool white fluorescent light.

The roots developed at the in vitro shoot-tip explants were separated and cut into smaller segments (1–1.5 cm long). Root segments were transferred in the laminar air flow hood into 250 mL Erlenmeyer flasks, filled with 100 mL of MS liquid medium supplemented with 5.4 μM α-naphthaleneacetic acid (NAA), 0.5 μM kinetin, and 30 g/L sucrose (pH: 5.8). It has been reported that the combined application NAA + kinetin helps in the production of saponins, flavonoids, and polyphenolic compounds [24–26]. The liquid root cultures were placed on a continuous rotary shaker (120 rpm) in a growth chamber (24 h dark, $22 \pm 1$ °C). After 30 days of culture, the new secondary adventitious roots formed over the primary roots constituted the material for further experimentation.

### 2.2. Effect of Temperature on Root Biomass and Species-Specific Secondary Metabolites in Adventitious Root In Vitro Production

The effect of three different incubation temperatures (10, 15, and 22 °C) on biomass and species-specific secondary metabolites production was investigated. The liquid medium used was the same as stock liquid root cultures enriched with L-proline 100 mg/L. Explants

of 1–1.5 cm long, taken from the secondary adventitious roots developed in the stock liquid cultures secondary roots, were placed in 250 mL Erlenmeyer flasks containing 100 mL of liquid medium and cultured on a continuous rotary shaker at 120 rpm, 24 h dark. There are reports showing that the exogenous application of proline effectively stimulates phenolics (phenols and flavonoids) and saponins content [27–30] under temperature stress [31]. The promotive role of exogenous application of L-proline on biomass growth and SMs production has been underlined in several reports [32–36] due to its osmolytic role in cells through alleviating arising stress during plant growth by elevating their levels of production and generation of higher or "over supply" of reducing equivalents [37].

The experiment included 3 treatments with 3 groups (Erlenmeyer flasks) of 30 explants. The initial fresh weight (FW) of inoculum (30 root segments) per flask was 1 g. After 60 days of culture, 15 different parameters were recorded. To produce root biomass (1) FW or root fresh biomass growth rate (=final FW/initial FW), (2) dry weight (DW), and (3) FW/DW ratio were assessed. For the production of secondary metabolites: (4) primulaverin, (5) primeverin, and (6) their cumulative content (primulaverin + primeverin) were determined as representative species-specific flavonoid glycosides. Among the triterpene saponins that have been reported in *P. veris,* (7) primulic acid I, (8) primulic acid II, and the (9) cumulative content of primulic acids (I + II) were assessed as the most predominant ones. All metabolites were expressed in mg/g DW. In an attempt to parametrize the combination efficacy of root biomass and production of bioactive compounds, the final root DW (g) was multiplied with the content in each targeted secondary metabolite and depicted as yield index (mg) (pure dry extract). As a result, the following six yield indexes (YI) were determined: (10) primulaverin YI, (11) primeverin YI, (12) (primulaverin + primeverin) YI, (13) primulic acid I YI, (14) primulic acid II YI, and (15) total primulic acids (I + II) YI.

*2.3. Comparison of Roots Secondary Metabolites Produced In Vivo Versus In Vitro*

Plants derived in vitro from rooted shoot tips gradually acclimatized and hardened to ex vitro conditions, firstly in the mist system of the greenhouse in 100 mL volume pots enriched with peat (Terrahum, Klasmann): perlite (Geoflor) (1:1 *v/v*), and afterwards in a net-house under 50% shading within 2.5 L pots consisted of peat moss (KTS2, Klasmann): perlite (Geoflor): soil substrate mixture (3:1:$\frac{1}{2}$ *v/v*) [22]. The KTS2 peat moss had a pH value of 6 and contained N-$P_2O_5$-$K_2O$ at a 14:16:18 ratio, respectively. The 1.5-year-old field grown plants did not receive any further fertilization. After blooming and fruit-set stages, roots from the 1.5-year-old in vivo plants were harvested in autumn. In particular, n = 15 in vivo root samples derived from 15 different plants were taken at the middle of October. The mean soil temperature in mid-October in 2.5 L depth pots containing in vivo plants was approximately 19 °C, as measured by the nearby Meteorological Station Observatory of the Aristotle University of Thessaloniki. The underground root system was separated from the aerial parts, washed with tap water to remove soil and debris, cut into smaller segments, and kept in −20 °C until further processing for analysis. The production of secondary metabolites in roots from in vitro culture (n = 3 samples/repetitions) and in vivo soil-grown plants (n = 15 samples/repetitions) (i.e., in vitro rooted plants fully acclimatized to the ex vitro environment after a 1.5-year growing period) were compared.

*2.4. Extraction of Flavonoids and Triterpene Saponins*

The root biomass obtained from in vitro cultures and ex vitro plants was washed-off from growing media (soil or turf), freeze-dried, and powdered using a laboratory Mill IKA A11. A quantity of 100 mg of milled tissue was mixed with 8 mL 80% methanol for the extraction of the flavonoids and triterpene saponins. The samples and solvent were mixed by orbital shaker for 3 h at room temperature, sonicated for 30 min, and the extraction proceeded overnight at 4 °C in the dark. Each extract was filtered on an MILLEX 13 mm-0.22 μm PTFE membrane into glass vial and the extracts were directly injected after extraction for LC-MS/MS analysis [38].

### 2.5. Ultra Performance Liquid Chromatography–Mass Spectrometry (UPLC-MS/MS, MRM) Analyses

Targeted UPLC analysis was performed on a Waters Acquity UPLC system (Milford, MA, USA) consisting of a binary pump, an online vacuum degasser, an autosampler, and a column compartment. Separation of the two flavonoid compounds (primeverin and primulaverin), and two main triterpene saponins (primulic acid I and II) was conducted on a Waters Acquity HSS T3 column 1.8 μm, 100 mm × 2.1 mm, and kept at 40 °C. The analysis was performed as described previously by Vrhovsek et al. [39] using water and acetonitrile as mobile phases for the gradient (Table 1).

Mass spectrometry detection was performed on a Waters Xevo TQMS instrument equipped with an electrospray (ESI) source. Data processing was performed using the Mass Lynx Target Lynx Application Manager (Waters). Capillary voltage was 3.5 kV in positive mode and −2.5 kV in negative mode; the source was kept at 150 °C; the desolvation temperature was 500 °C; cone gas flow was 50 L/h; and desolvation gas flow was 800 L/h. Unit resolution was applied to each quadrupole. Flow injections of each individual metabolite were used to optimize the MRM conditions. Data processing was performed using the Mass Lynx Target Lynx Application Manager (Waters) (Table 1).

**Table 1.** Multiple reaction monitoring (MRM) parameters of primeverin, primulaverin, primulic acid I and II, inserted to the method of Vrhovsek et al. [39].

| | | | | | Quantifier Ion | | Qualifier Ion | |
| --- | --- | --- | --- | --- | --- | --- | --- | --- |
| Compound | Rt (min) | ES | Cone Voltage (V) | Q1 (*m/z*) | Collision Energy (V) | Q2 (*m/z*) | Collision Energy (V) | Q2 (*m/z*) |
| Primeverin | 4.1 | - | 22 | 475.2234 | 6 | 293.076 | 24 | 181.027 |
| Primulaverin | 3.82 | - | 22 | 475.16 | 6 | 293.08 | 44 | 166.013 |
| Primulic acid I | 10.23 | - | 80 | 1103.649 | 58 | 455.412 | 38 | 205.069 |
| Primulic acid II | 10.01 | - | 84 | 1235.713 | 64 | 100.947 | 78 | 455.403 |

### 2.6. Statistical Analysis

The experimental layout was completely randomized. The means were subjected to analysis of variance (one-way ANOVA) and compared using the Duncan multiple-range test ($p < 0.05$) using the statistical program SPSS 17.0 (SPSS Inc., Chicago, IL, USA). The in vitro experiment included three treatments (three temperatures: 10, 15, 22 °C as independent variable) with three repetitions (three adventitious root samples or repetitions) per treatment for root biomass and species-specific secondary metabolites' production (dependent variables).

In the case of the roots harvested from the in vivo soil-grown plants (i.e., in vitro rooted plants fully acclimatized to the ex vitro environment after a 1.5-year growing period), n = 15 roots samples (repetitions) were taken from 15 different plants and used for biomass and species-specific secondary metabolites production.

The means for dry biomass and secondary metabolites production (dependent variables) of the in vivo roots were further compared based on their absolute values (non-statistical analysis conducted) with those of the in vitro roots incubated under the highest temperature of 22 °C (in vivo vs. in vitro—22 °C, independent variable).

### 3. Results

#### 3.1. Effect of Temperature on In Vitro Development and in Species-Specific Secondary Metabolites' Production

All parameters evaluated were positively affected under the highest incubation temperature. In detail, fresh weight (6.04 g) or root biomass growth rate (×6.04), primulic acid I content (6.09 mg/g DW), cumulative saponins content (primulic acid I + primulic acid II, 6.60 mg/g DW), primulic acid I YI (6.28 mg), and cumulative saponins (primulic acids I + II) YI (6.87 mg) were significantly higher under the highest incubation temperature of 22 °C. For all these parameters, there was a positive correlation between temperature increase (10, 15, 22 °C) and biomass and/or secondary metabolites production. On the

other hand, root FW/DW ratio (5.10–6.05 g), primulaverin (8.70–9.93 mg/g DW), and total flavonoid glycosides (primulaverin + primeverin) (8.73–10.03 mg/g DW) accumulation were not affected significantly by the temperature increase, as revealed by the statistical analysis. However, primulaverin YI (7.19 and 10.04 mg), primeverin YI (0.07 and 0.09 mg), and cumulative flavonoid glycosides YI (7.26 and 10.13 mg) under 15 °C and 22 °C were significantly higher as compared to 10 °C (4.09 mg, 0.01 mg, and 4.10 mg in primulaverin YI, primeverin YI, and cumulative flavonoids YI, respectively). To be specific, primulaverin YI and cumulative flavonoid glycosides YI under 22 °C were 2.5-fold higher than under 10 °C, while primeverin YI at 22 °C was 9 times higher as compared to 10 °C. Root DW was significantly higher under 22 °C and lower under 10 °C, whereas the intermediate temperature of 15 °C represented values without significant difference to the other two. Therefore, taking simultaneously into consideration all biomass and bioactive compounds production, as well as the relative yield indexes, 22 °C was revealed to be the optimum temperature for improved performance concerning the adventitious root cultures of *P. veris* subsp. *Veris* in liquid medium enriched with 100 mg/L L-proline (Figure 2, Table 2).

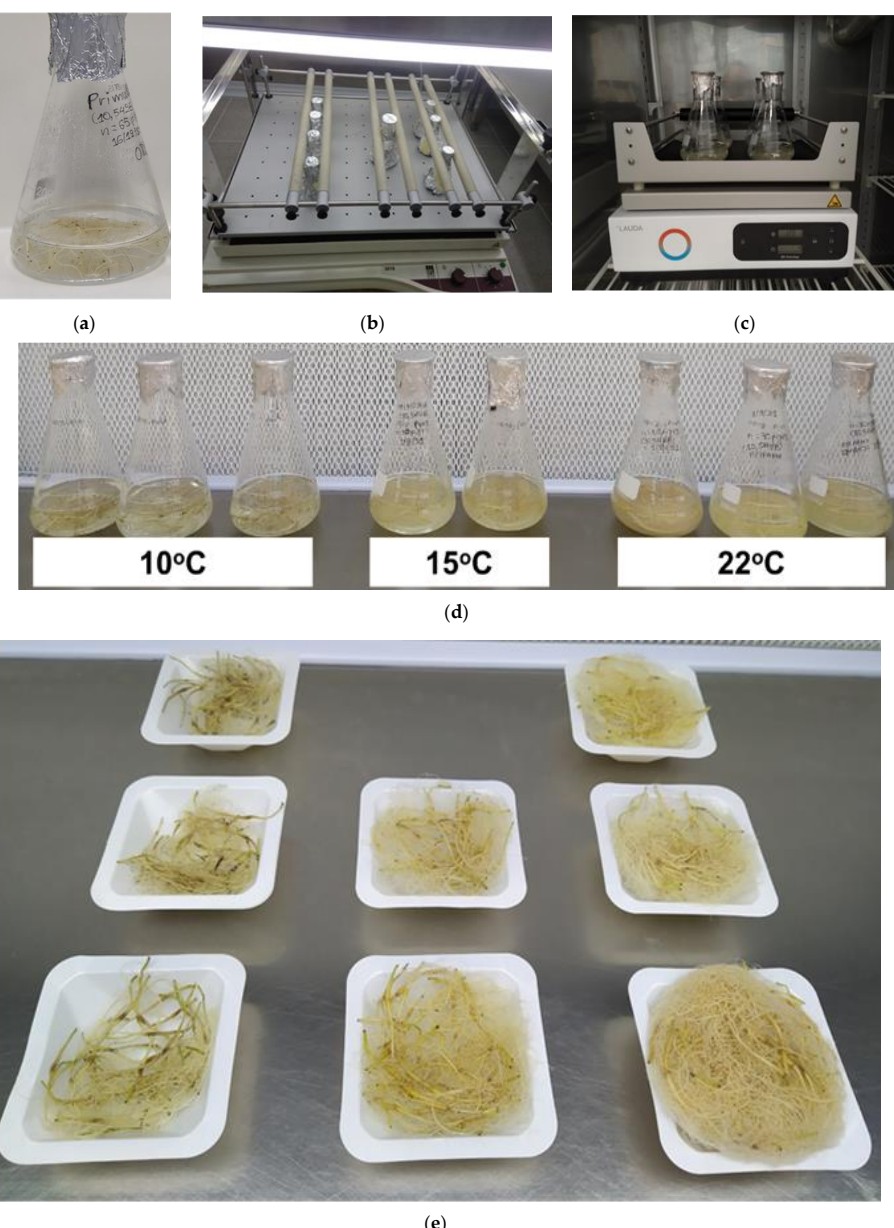

**Figure 2.** (**a**) Adventitious in vitro roots of *Primula veris* subsp. *veris* cultured in 250 mL Erlenmeyer flasks containing 100 mL of liquid MS medium enriched with 5.4 μM NAA, 0.5 μM kinetin, 100 mg/L

L-proline, and 30 g/L sucrose (pH: 5.8) at the initiation of the experiment on day 0; (**b,c**) In vitro roots agitated on continuous rotary shakers (120 rpm) under 24 h dark; (**d,e**) Biomass growth and species-specific secondary metabolites' production of adventitious roots after 60 days of in vitro culture under three different temperatures (10, 15, 22 °C) inside and outside flasks, respectively.

**Table 2.** Effect of temperature (10, 15, 22 °C) in adventitious root culture on biomass and species-specific methylated flavonoid glycosides and saponins production after 60 days of in vitro culture as well as biomass and species-specific secondary metabolites production in the roots of 1.5-year-old in vivo plants of *Primula veris* subsp. *veris* grown in the soil.

| Secondary Metabolites | Content (mg/g DW) and Yield (mg/100 mL Medium/250 mL Flask) In Vitro | | | | Content (mg/g DW) and Yield (mg/2.5 L Pot) In Vivo |
|---|---|---|---|---|---|
| | $T_{10}$ | $T_{15}$ | $T_{22}$ | *p*-Value | |
| Primulaverin content | 8.70 ± 0.83 a | 9.93 ± 0.66 a | 9.71 ± 0.38 a | 0.420 ns | 10.23 ± 0.80 |
| Primeverin content | 0.03 ± 0.02 b | 0.10 ± 0.01 a | 0.09 ± 0.01 a | 0.048 * | 0.28 ± 0.03 |
| Cumulative flavonoids | 8.73 ± 0.85 a | 10.03 ± 0.67 a | 9.80 ± 0.38 a | 0.398 ns | 10.51 ± 0.82 |
| Primulic acid I content | 2.85 ± 0.31 c | 4.14 ± 0.05 b | 6.09 ± 0.20 a | 0.000 *** | 17.01 ± 0.36 |
| Primulic acid II content | 0.20 ± 0.03 b | 0.53 ± 0.01 a | 0.51 ± 0.06 a | 0.045 * | 0.09 ± 0.01 |
| Cumulative saponins content | 3.05 ± 0.33 c | 4.67 ± 0.03 b | 6.60 ± 0.40 a | 0.000 *** | 17.10 ± 0.52 |
| Primulaverin yield | 4.09 ± 0.51 c | 7.19 ± 0.76 ab | 10.04 ± 2.19 a | 0.049 * | 33.55 ± 2.85 |
| Primeverin yield | 0.01 ± 0.01 b | 0.07 ± 0.01 a | 0.09 ± 0.02 a | 0.018 * | 0.92 ± 0.13 |
| Cumulative flavonoids yield | 4.10 ± 0.52 c | 7.26 ± 0.77 ab | 10.13 ± 2.21 a | 0.038 * | 34.47 ± 0.67 |
| Primulic acid I yield | 1.33 ± 0.14 c | 2.98 ± 0.09 b | 6.28 ± 1.31 a | 0.010 * | 55.79 ± 1.27 |
| Primulic acid II yield | 0.10 ± 0.02 b | 0.38 ± 0.02 a | 0.59 ± 0.38 a | 0.045 * | 0.30 ± 0.02 |
| Cumulative saponins yield | 1.42 ± 0.15 c | 3.36 ± 0.11 b | 6.87 ± 1.66 a | 0.019 * | 56.09 ± 1.96 |
| **Biomass** | **In Vitro** | | | | **In Vivo** |
| | $T_{10}$ | $T_{15}$ | $T_{22}$ | *p*-Value | |
| Total fresh weight (g) | 2.38 ± 0.08 c | 4.09 ± 0.10 b | 6.04 ± 0.73 a | 0.003 ** | 18.40 ± 0.81 |
| Total dry weight (g) | 0.47 ± 0.02 b | 0.72 ± 0.03 ab | 1.02 ± 0.18 a | 0.029 * | 3.28 ± 0.13 |
| Fresh weight/dry weight ratio | 5.10 ± 0.08 a | 5.68 ± 0.08 a | 6.05 ± 0.46 a | 0.123 ns | 5.61 ± 0.02 |

Cumulative flavonoids: Primulaverin + Primeverin; Cumulative saponins: Primulic acid I + Primulic acid II; In vitro adventitious roots: Means (n = 3 samples or 3 flasks × 25 explants/flask/treatment) ± standard error (S.E.) with the same letter in a row regardless temperature are not statistically significant different from each other according to the Duncan's multiple range test at $p \leq 0.05$. ns $p > 0.05$; * $p \leq 0.05$; ** $p \leq 0.01$; *** $p \leq 0.001$. In vivo roots derived from 1.5-year-old soil-grown plants: Means (n = 15 root samples) ± standard error (S.E.) in a row are provided for each parameter evaluated.

### 3.2. Comparison of Species-Specific Secondary Metabolites of Roots Produced In Vivo and In Vitro

In order to elucidate whether in vitro root culture under 22 °C could be an alternative potential and sustainable system for the production of *P. veris* root biomass with high SMs production compared to conventional roots of 1.5-year-old in vivo plants grown in the soil, the content and yield index (YI) of species-specific methylated flavonoid glycosides (primulaverin, primeverin, primulaverin + primeverin) and saponins (primulic acids I, II, I + II) as well as the biomass parameters (absolute values) were also recorded. Primulic acid II exhibited almost three-fold higher content (0.51 mg/g DW) and two-fold higher YI (0.59 mg/100 mL liquid medium/250 mL flask) in the in vitro-produced root raw material compared to the soil-grown roots (0.09 mg/g DW and 0.30 mg/2.5 L pot). In contrast, the fresh biomass growth rate (×18.4), dry weight (3.28 g), primeverin (0.28 mg/g DW), and primulic acid I content (17.01 mg/g DW) in the in vivo roots were approximately three-fold higher than in the in vitro roots under 22 °C, being ×6.04, 1.02 g, 0.09 mg/g DW, and 6.09 mg/g DW, respectively. In addition, YIs (mg/2.5 L pot) of primulverin (33.55 mg), primeverin (0.92 mg), cumulative flavonoids (34.47 mg), primulic acid I (55.79 mg), and cumulative saponins (56.09 mg) were 3.3, 10.2, 3.4, 8.9, and 8.2 times higher in the in vivo roots as compared to the in vitro roots, being 10.04, 0.09, 10.13, 6.28, and 6.87 mg/100 mL liquid medium/250 mL flask, respectively. On the other hand, the primulaverin (10.23 mg/g DW)

and cumulative flavonoids (10.51 mg/g DW) content in the in vivo roots were at similar levels to the in vitro roots, being 9.71 mg/g DW and 9.80 mg/g DW, respectively, whereas cumulative saponins content (10.51 mg/g DW) in the in vivo roots was 1.6-fold higher than in the in vitro roots (6.60 mg/g DW). Root biomass and SMs production (content and YI) was higher in the in vivo roots than in the in vitro roots after 60 days of culture, except for primulic acid II (Table 2).

## 4. Discussion

It is well documented that a variety of biotic and abiotic factors may influence the production of secondary metabolites *in planta*, either in conventional soil-grown crops or under in vitro culture conditions. Besides other advantages, an in vitro system for the production of secondary metabolites serves as a sustainable technique that maintains the genetic stability, limits the uncontrolled collection of threatened species from its natural environment, ensures the repeatable biomass production of a highly sustainable quality independent of environmental conditions, ensures that the biomass produced is free of foreign substances used in crop systems (pesticides, weeds, soil particles etc.), and therefore minimizes the postharvest treatments prior to extraction processing and isolation (especially for root-cultures) [40].

In the extracts of the roots, leaves, and flowers of *P. veris*, various polyphenolic compounds have been identified alongside most characteristic ones such as primeverin, primulaverin, catechin, astragalin, chlorogenic acid, rutin, kaempferol, hydroxy-dimethoxyflavone, and quercetin-3-*O*-dihexoside [9,41]. Primulaverin and primeverin degrade during storage in the presence of the enzyme primverase, giving birth to the typical fragrance of the drug [42], thus, they not only serve as marker compounds but also as indicators of the age of the plant material [9,11]. Apart from polyphenols, cowslip is also characterized by some species-specific saponins, with the most predominant ones being primulic acid I and II, the concentrations of which were observed to be higher in roots determined through HPLC analysis [11,43].

In nature, the production of SMs exists in small amounts. The current supply of saponins (in the international trade market) extracted from plants grown conventionally in the field is generally considered a laborious and low yielding process. This, together with cultivation-focused problems of *P. veris* crops [13,44], has resulted in the exploitation of different biotechnological systems for root biomass production similar to other plant species such as *Datura stramonium* and *Hyoscyamus muticus* [45], *Atropa belladonna* [45,46], and *Panax ginseng* [21], among others. In addition, the collection of cowslip raw materials from natural resources does not guarantee high quality, as concentrations of the biologically active compounds, similarly to many other medicinal plants, very often may vary [47].

Considering that *Primula* species thrive in subalpine zones with a general optimum temperature for plant development of 15 °C [48], a species-specific temperature experiment was performed to identify the necessary conditions for higher SMs production in adventitious root cultures. According to our data, an induction of the root biomass production was observed together with individual and cumulative flavonoids and saponins (which also resulted in increased yield indexes), under the effects of increased temperatures, from 10 to 15 and 22 °C. Such changes could be attributed to the direct influence of the media temperature on the roots' physiology through variations on their rate intracellular reactions, and as a result their variable metabolic regulation [49]. Furthermore, our findings agree with Jochum et al. [50], who reported an increment of secondary metabolite concentrations in *Panax quinquefolius* under elevated temperatures of culture conditions. In particular, plants grown at a higher temperature (31.2 °C day/26.6 °C night) had significantly less belowground biomass and higher concentrations of total storage root ginsenosides (i.e., triterpenoid saponins) than plants grown at a lower temperature (26.8 °C day/21.2 °C night), thus *Panax quinquefolius* was highly sensitive to a 5 °C increase in growth temperature [50].

Previous studies on 'rooty' cultures comparing SMs production between in vitro and in vivo growth systems have reported both induction and reduction to their content in

plant tissues. For example, in vitro grown hairy roots from *Atropa belladonna* revealed higher scopolamine and atropine content (0.024 and 0.371% on a DW basis, respectively), as compared with one-year-old field-grown plants, which contained 0.008 and 0.34% DW, respectively [45]. Furthermore, adventitious roots of *Hypericum perforatum* produced higher levels of phenols, flavonoids, hypericins, and chlorogenic acid than when cultivated in flasks and rapidly produced root clumps with continuously increasing biomass throughout the culture period in a 3-L balloon type fermenter [51]. On the other hand, the average content of primulic acid I of *P. veris* roots from in vitro micropropagated plants and adventitious root cultures was 1.5 to two times lower than in roots of soil-grown plants, whereas in callus and suspension cell cultures the content was eight times lower [52]. The data presented herein are in accordance with the report of Okrslar et al. [52], which considered the lower reported primulic acid I content in in vitro adventitious roots as compared to the soil-grown roots' biomass. However, such differences could be also attributed to the different ontogenetic stage of the root tissue, considering that the in vitro adventitious roots were preserved in a relative juvenile developmental stage, while the soil-grown roots were harvested from mature plants (1.5-year-old), and after fruit-set when plants usually reallocate most of their energy sources to the underground organs in order to scope with the biotic factors during the autumn and winter period.

In the present study with *P. veris* subsp. *veris*, the total culture period for the in vivo plants prior to harvesting the roots was one and a half years or 18 months (540 days) while for the in vitro plants the total culture period was only two months (60 days); therefore, nine times more time is needed to obtain these values for the production of biomass and species-specific bioactive compounds in the in vivo roots than in the in vitro ones. In addition, the YI of species-specific SMs in the in vivo vs. the in vitro roots are expressed in different volumes and culture periods, in specific as mg/2.5 L pot within 540 days in the case of in vivo roots and as mg/100 mL liquid medium/250 mL flask within 60 days in the case of in vitro roots. From the results, it is evident that higher biomass and species-specific SMs production (except of primulic acid II), but in a longer growing period by nine times, are obtained in the in vivo harvested roots from soil-grown plants in comparison to the in vitro roots, indicating that as well as the maximum biomass and production of bioactive compounds, the duration of the culture period plays a significant role that cannot be neglected or underestimated, as well as their combined effect as a single factor. Indeed, results for biomass production and yield index of SMs in the in vitro roots after 540 days of culture ($\times$ nine times) instead of 60 days as presented herein would be different, probably higher even, from those achieved in the case of the roots of the 1.5-year (18 months, 540 days) in vivo soil-grown plants reported in this study. Therefore, the comparison of cumulative flavonoid glycosides and saponins content between the soil-grown roots and the in vitro adventitious root cultures of *P. veris* subsp. *veris* within the same cultivation period of 1.5 years showed that the production of adventitious roots in vitro investigated in our study could serve as biotechnological 'factories' that could host the efficient production of secondary metabolites in *P. veris* subsp. *veris* roots in similar way to other 'rooty' cultures. This can be supported by the hypothesis that plants from conventional field crops are harvested only once every two years for the valorization of their roots, while during in vitro culture it is possible to end up with a much higher crude extract yield index due to higher root biomass production from repeatable cultures. The fresh and dry biomass production, primeverin, and primulic acid I content in the in vivo roots were approximately three-fold higher than in the in vitro roots under 22 °C. In addition, the yield indexes (mg/2.5 L pot) of primulverin, primeverin, cumulative flavonoids, primulic acid I, and cumulative saponins were 3.3, 10.2, 3.4, 8.9, and 8.2 times higher in the in vivo roots as compared to the in vitro roots under 22 °C, respectively. It is evident therefore that root biomass and SMs production (content and YI) were higher in in vivo roots from 1.5-year-old soil-grown plants than in in vitro roots after 60 days of culture, except for primulic acid II. In particular, the content and yield index of in vitro 60-day cultured roots in primulic acid II were three-fold and two-fold higher, respectively, than in in vivo roots

from 1.5-year-old soil-grown plants. In addition, from a post-harvest point of view, in vitro culture-obtained primula root biomass is considered much easier and cost effective in processing raw material, considering also the cost of post-harvest treatments in order to end up with clean root biomass raw material (without soil compartments and/or separated from aerial parts of the plants) for further extraction processes [40]. As a result, the stable production of a specific quality of primula root crude extract, independent from annual environmental and biotic factors affecting the plant, could result in more constant formulas that would be easier to exploit in end-products for the industry. Taking into consideration that flavonoid glycosides and saponins production could be increased by adding elicitors into the liquid media, it appears that the vast potentiality of this species' adventitious root culture could be a sustainable and good source towards the exploitation of the scaling up production of biologically active substances like flavonoids and saponins.

## 5. Conclusions

Overall, this study revealed the potential sustainable production of root biomass of *P. veris* in vitro towards the extraction of valuable triterpene saponins and flavonoids for the pharmaceutical industry. The optimum temperature that ensures the root development in vitro was identified at 22 °C, which promoted root fresh and dry weight of about 50% comparing to root weight developed under 10 °C. Furthermore, it seems that temperature affects the accumulation of triterpene saponins in adventitious roots of *P. veris* developed under the tested in vitro system, with the temperature of 22 °C expressing almost two-fold higher content in both primulic acids (I and II). This study could help set the basis for the further evaluation of putative elicitors (hormonal, chemical a.o) that could promote further the development of root biomass in vitro and, in parallel, the biosynthesis of these species-specific secondary metabolites in large scale in bioreactors.

**Author Contributions:** Conceptualization, K.G..; methodology, V.S., E.S. and K.G.; software, V.S., E.S., A.A. and S.M.; validation, K.G.; formal analysis, V.S., E.S., A.A. and S.M.; investigation, V.S., E.S. and A.A; resources, S.M., E.M. and K.G.; data curation, V.S., E.S. and A.A.; writing—original draft preparation, V.S., E.S. and K.G.; writing—review and editing, V.S., E.S. and K.G.; visualization, V.S., E.S. and K.G; supervision, K.G.; project administration, K.G.; funding acquisition, K.G. All authors have read and agreed to the published version of the manuscript.

**Funding:** This research was co-financed by the European Regional Development Fund of the European Union and Greek national funds through the Operational Program Competitiveness, Entrepreneurship and Innovation, under the call RESEARCH–CREATE–INNOVATE (project code: T2EΔK-02927, MIS 5069915) entitled "Development and optimization of in vitro culture methods in bioreactors to produce repeatable and excellent quality plant material for extracts used in food supplements and cosmetics" (Acronym: BIOREACT).

**Institutional Review Board Statement:** Not applicable.

**Informed Consent Statement:** Not applicable.

**Data Availability Statement:** Not applicable.

**Conflicts of Interest:** The authors declare no conflict of interest.

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
