# Peer review of "Species-Specific Secondary Metabolites from Primula veris subsp. veris Obtained In Vitro Adventitious Root Cultures: An Alternative for Sustainable Production"

_sustainability, doi:10.3390/su15032452_

Round 1

Reviewer 1 Report

Dear author(s),

there are some inspiring insights thorough the manuscript and I tend to agree on its publication. However, there are few points that needs to be quickly addressed to improve its overall communication:

Title:

1/ clearly condensate the environmental novelty and industrial significance of the main discovery into a short and groundbreaking claim

Abstract:

2/ better explain what the research hypothesis is and why its confirmation or refutation is urgent in terms of "sustainability"

3/ avoid the use of abbreviations, jargon and technical terms, please understand that the purpose of the Abstract is to explain to all readers (including those from other disciplines) what the paper is about

4/ clearly indicate how will our international audience benefit from these revelations (higher level of generalization is needed)

Introduction:

5/ do not provide our readers with encyclopaedic information about Primula veris, rather explain how is this research relates to financial or environmental sustainability

6/ justify the industrial significant significance by providing corresponding financial numbers, refer to papers that deal with the sustainability-manufacturing nexus such as "Sustainable Organizational Performance, Cyber-Physical Production Networks, and Deep Learning-assisted Smart Process Planning in Industry 4.0-based Manufacturing Systems" and "Artificial Intelligence Data-driven Internet of Things Systems, Real-Time Advanced Analytics, and Cyber-Physical Production Networks in Sustainable Smart Manufacturing"

7/ the economy of production and processing should not be ignored

8/ go straight to the point and more in depth, write more technically (always provide corresponding numbers), significantly condensate all the text by reducing ballast phrases and cliché

9/ the research hypothesis could be stated more clearly, condensate the research hypothesis into 1 short statement or question that will be subsequently confirmed or refuted, make sure the urgency and significance of the research hypothesis was justified in its environmental - economic nexus

Materials and Methods:

10/ the method must be presented in such a way that it can be reproduced anytime, by anyone, anywhere (do not create obstacles like referring to specific location etc.)

11/ provide cost breakdown or at least some simplified financial analysis if you are about to argue that this concept is sustainable (realistic)

Results:

12/ each Tab. and Fig. should be provided with caption that describes A/ what can be seen and B/ how is this relevant to the research hypothesis

13/ avoid data overkill, present only the most most industrially important results

Discussion:

14/ show more self-criticism to your work (can all the methods and results be fully trusted? what are the weaknesses of the methods used? where do the main inaccuracies arise? what are the limitations from a commercial point of view? are the lessons learned transferable to other fields?)

15/ deeper review the latest trends in plant production, refer to papers "Silica nanoparticles from coir pith synthesized by acidic sol-gel method improve germination economics" and "Techno-economic analysis reveals the untapped potential of wood biochar"

16/ compare your results in more depth with the existing literature, identify the main deviations and try to explain the mechanisms by which they may have been caused

17/ reveal the main driving mechanisms of your results, provide deeper synthesis and reveal some more original/significant findings

Conclusions:

18/ the most important chapter is hard to find, clearly indicate whether the research hypotheses tends to be confirmed or not

19/ present only original and industrially significant revelations that have the potential to expand the horizon of human knowledge (higher level of generalization is mandatory)

Author Response

Dear Reviewer #1,

We are grateful to receive reviews that help us improve the quality of our manuscript. Following the reviewer #1 comments we corrected the text adopting most of the comments suggested. Please find below our comments to the review. In the revised manuscript, all changes and new additions in text and throughout the manuscript are highlighted with letters of blue color. In addition, in this response letter to your comments, the authors gave responses below each comment in blue color lettering format.

Reviewer #1

Comments and Suggestions for Authors

Dear author(s),

there are some inspiring insights thorough the manuscript and I tend to agree on its publication. However, there are few points that needs to be quickly addressed to improve its overall communication:

Title:

1/ clearly condensate the environmental novelty and industrial significance of the main discovery into a short and groundbreaking claim- We revised the title making clearer the sustainable point of view of this manuscript

Abstract:

2/ better explain what the research hypothesis is and why its confirmation or refutation is urgent in terms of "sustainability"

- We believe that is clear how the hypothesis of the present work is interrelated to sustainability especial environmental and as an industrial production system of plant secondary metabolites. In fact, the last sentence has been changed according to the comments of all reviewers.

3/ avoid the use of abbreviations, jargon and technical terms, please understand that the purpose of the Abstract is to explain to all readers (including those from other disciplines) what the paper is about-corrected

4/ clearly indicate how will our international audience benefit from these revelations (higher level of generalization is needed)

 -Not clear what the reviewer wants exactly. In any way we included in the end of the manuscript a ‘Conclusion’ part where some aspects are now discussed about the benefits of such production systems.

Introduction:

5/ do not provide our readers with encyclopaedic information about Primula veris, rather explain how is this research relates to financial or environmental sustainability

  • So far each Journal is common to have a ‘targeted’ and quite specific scientific audience. In most of the Journals we published so far and generally is recommended to structure the Introduction part started from general information given downgrading to specific which is the aim of the work presented. Nevertheless, we included an additional paragraph early to the introduction part on the relation of P. veris and environmental sustainability.

6/ justify the industrial significant significance by providing corresponding financial numbers, refer to papers that deal with the sustainability-manufacturing nexus such as "Sustainable Organizational Performance, Cyber-Physical Production Networks, and Deep Learning-assisted Smart Process Planning in Industry 4.0-based Manufacturing Systems" and "Artificial Intelligence Data-driven Internet of Things Systems, Real-Time Advanced Analytics, and Cyber-Physical Production Networks in Sustainable Smart Manufacturing"

  • The financial part was out of the scope of the present study and is out of the co-authors expertise to develop such a financial study. Opening the suggested papers from the reviewer we couldn’t find any connection to the topic of our study to these papers. They are irrelevant to the our topic and therefore we can not include financial part within this manuscript as it could be a different manuscript on itself.

7/ the economy of production and processing should not be ignored

- Of course the economy of production and processing of secondary metabolites is very important but was not the topic of the present study. It is the first time that the production in vitro of these species specific-secondary metabolites referred in a scientific paper, and this was the aim of our study.

8/ go straight to the point and more in depth, write more technically (always provide corresponding numbers), significantly condensate all the text by reducing ballast phrases and cliché

-the comment is to general and doesn’t refer to specific line/page as the rest of the reviews and we couldn’t understand which parts should be improved. Please if there are specific lines let us know.

9/ the research hypothesis could be stated more clearly, condensate the research hypothesis into 1 short statement or question that will be subsequently confirmed or refuted, make sure the urgency and significance of the research hypothesis was justified in its environmental - economic nexus

The aims of the present work were corrected referring also to the environmental sustainability. The economic nexus is out of scope for this study.

Materials and Methods:

10/ the method must be presented in such a way that it can be reproduced anytime, by anyone, anywhere (do not create obstacles like referring to specific location etc.)

Such information on the locations that plant material obtained is always requested in such studies involving medicinal plants. In fact they are not given to create obstacles but to describe better the plant material used for each study which is important for several reasons. It should be noticed that a number of factors affecting the SMs of plants including genotype, environmental a.o. Here we clearly give information about the genotypes used for our study and this is important to be shown.

11/ provide cost breakdown or at least some simplified financial analysis if you are about to argue that this concept is sustainable (realistic)

As mentioned before the co-authors have no expertise on agricultural economics and therefore is impossible to provide the requested information. Also it is out of scope of the present study that deals with plant science and in vitro culture referring that affect indirectly the environmental sustainability. The requested information is completely another topic from the one presented herein.

Results:

12/ each Tab. and Fig. should be provided with caption that describes A/ what can be seen and B/ how is this relevant to the research hypothesis

-corrected

13/ avoid data overkill, present only the most most industrially important results

-is not clear which data the reviewer means. Please refer to specific lines/pages of the text so that we can correct.

Discussion:

14/ show more self-criticism to your work (can all the methods and results be fully trusted? what are the weaknesses of the methods used? where do the main inaccuracies arise? what are the limitations from a commercial point of view? are the lessons learned transferable to other fields?)

Corrected according to also others reviewers’ comments.

15/ deeper review the latest trends in plant production, refer to papers "Silica nanoparticles from coir pith synthesized by acidic sol-gel method improve germination economics" and "Techno-economic analysis reveals the untapped potential of wood biochar"

These topics are irrelevant with ours.

16/ compare your results in more depth with the existing literature, identify the main deviations and try to explain the mechanisms by which they may have been caused

Corrected also according to others reviewers’ comments.

17/ reveal the main driving mechanisms of your results, provide deeper synthesis and reveal some more original/significant findings

 -The reviewer insists requesting discussion parts that are relevant to economics and financials which are out of the scope of the present study. Generally all comments referring to discussion part are very general therefore we adopted the comments of the rest 3 reviewers to improve this part.

Conclusions:

18/ the most important chapter is hard to find, clearly indicate whether the research hypotheses tends to be confirmed or not and 19/ present only original and industrially significant revelations that have the potential to expand the horizon of human knowledge (higher level of generalization is mandatory)

-A conclusion part was included to the manuscript summarizing the most important findings of this work. The conclusion part has been changed according also to others reviewers’ comments.

Reviewer 2 Report

The authors of this manuscript determined two flavonoids, and two triterpene saponins produced in vitro from Primula veris adventitious root cultures.

The manuscript in general was well written and designed; however, there are some points that should be clarified.

  1. Line 22: triterpene saponins.
  2. The abstract is descriptive. It would be better if the authors give some values of these secondary metabolites produced in the cultured and soil-growing plants.
  3. The keywords are too many and should be reduced.
  4. Line 42: what is primrose, please use either scientific or common name.
  5. Line 46: influence.
  6. The authors did not explain if the field grown plants receive any fertilization or any chemicals during their growth (1.5 year) that could affect the concentrations of its secondary metabolites. This could affect the overall results of the study.
  7. Line 154: triterpene saponins.
  8. Line 205: Please identify at which temperature root biomass that you compare with field grown plants.
  9. Line 206: P. veris should be italic.
  10. Line 2013: there is no statistical analysis in table 3 to be used in the comparison between the two samples.
  11. Lines 241-242: delete (Kamada et 241 al. 1986; Palazon et al. 2008).
  12. Line 303: give the full name of H. perforatum.
  13. The discussion should be improved. The authors should discuss the effect of media components (NAA, kinetin, and proline) on the plant compounds.

Author Response

Dear Reviewer #2,

We are grateful to receive reviews that help us improve the quality of our manuscript. Following the reviewer #2 comments we corrected the text adopting most of the comments suggested. Please find below our comments to the review. In the revised manuscript, all changes and new additions in text and throughout the manuscript are highlighted with letters of blue color. In addition, in this response letter to your comments, the authors gave responses below each comment in blue color lettering format.

Reviewer #2

Comments and Suggestions for Authors

The authors of this manuscript determined two flavonoids, and two triterpene saponins produced in vitro from Primula veris adventitious root cultures.

The manuscript in general was well written and designed; however, there are some points that should be clarified.

 Line 22: triterpene saponins.

- Corrected, “triterpene saponins” instead of “triterpene saponines”

  1. The abstract is descriptive. It would be better if the authors give some values of these secondary metabolites produced in the cultured and soil-growing plants.

- In the revised version, we provided in the abstract section, the values of species-specific secondary metabolites (primulaverin, primeverin, primulic acid I, primulic acid II)

  1. The keywords are too many and should be reduced.

- We reduced the number of keywords.

  1. Line 42: what is primrose, please use either scientific or common name.

- We substitute the term with the common name of the studied plant species which is “cowslip”

  1. Line 46: influence.

-  Replaced as suggested.

  1. The authors did not explain if the field grown plants receive any fertilization or any chemicals during their growth (1.5 year) that could affect the concentrations of its secondary metabolites. This could affect the overall results of the study.
  • Field grown plants did not receive any fertilization, different from what the peat substrate mixture (KTS2, Klasmann) had (pH 6, N-P-K at a 14:16:18 ratio, respectively). Details were added at Materials and Methods part. Besides the aim of the study was the in vitro production of secondary metabolites.
  1. Line 154: triterpene saponins.

- In the revised manuscript, “triterpene saponines” was replaced by “triterpene saponins” as suggested.

  1. Line 205: Please identify at which temperature root biomass that you compare with field grown plants.

- In the revised manuscript, we identify that comparison of all parameters were done at 22oC

  1. Line 206: P. veris should be italic.

- Corrected.

  1. Line 2013: there is no statistical analysis in table 3 to be used in the comparison between the two samples.
  • The aim of the analysis of the soil-grown roots was to detect if the same species-specific secondary metabolites exist and in which amount approximately as an indicator for in vitro produced compounds. Tables 2 and 3 has been merged, but only for making easier for readers. Statistical analysis cannot be merged, because all grown parameters are different (in vitro-soil, duration of culture, different temperatures, root culture – whole plant etc).
  1. Lines 241-242: delete (Kamada et 241 al. 1986; Palazon et al. 2008).

- Deleted both from text as required and from the reference list, thus numbering order of citations in the reference list based on their appearance order in the text was re-arranged.

  1. Line 303: give the full name of  perforatum.

- The full name of the plant taxon was given, which is “Hypericum perforatum

  1. The discussion should be improved. The authors should discuss the effect of media components (NAA, kinetin, and proline) on the plant compounds.
  • It has been improved.

Reviewer 3 Report

The manuscript entitled 'Species-specific secondary metabolites production in vitro from Primula veris subsp. veris adventitious root cultures' authored by Sarropoulou et al. is very interseting and informative describing the effect of different temperatures on biomass production and alternative production of SMs through in vitro adventitious root of P. veris other than natural resources. The protocol described would be highly informative and interesting for researchers who work in the production of pharmaceutically important SMs from plants and helps to further standardize to increase the amounts. A few minor corrections are recommended before considering this manuscript for publication.

Minor comments

Line 82: It is not clear how the authors arrived at the concentrations and selection of hormones from the stock to subsequent in vitro root cultures (5.4 μM α-naphthaleneacetic acid (NAA), 0.5 μM kinetin)? Is it adopted from published literature like stock culture (16) or standardized by the authors earlier? Please mention.

Line 86: laminar air blow hood change to laminar air flow hood 

Line 90: 24h dark, T 22 ± 1°C temperature. What is T represented here?

Line 129: microshoots in vitro and gradually acclimatized... Change to microshoots in vitro was gradually acclimatized

Line 190: primeverin YI (0.09-0.10 mg). Check the values mentioned in Table 2

Line 191-192. parameters were almost 2- fold higher under 15 oC and 22 oC compared to 10 oC. It is not really almost 2-fold increase. Better check and change the statement accordingly.

Line 206: P. veris change to Italics

Line 214-216: It is interesting to see the soil-grown roots accumulated almost 3-fold higher content of primulic acid. Could you monitor the soil temperature or other parameters like pH that lead to this difference?

Author Response

Dear Reviewer #3,

We are grateful to receive reviews that help us improve the quality of our manuscript. Following the reviewer #3 comments we corrected the text adopting most of the comments suggested. Please find below our comments to the review. In the revised manuscript, all changes and new additions in text and throughout the manuscript are highlighted with letters of blue color. In addition, in this response letter to your comments, the authors gave responses below each comment in blue color lettering format.

Reviewer #3

Comments and Suggestions for Authors

The manuscript entitled 'Species-specific secondary metabolites production in vitro from Primula veris subsp. veris adventitious root cultures' authored by Sarropoulou et al. is very interesting and informative describing the effect of different temperatures on biomass production and alternative production of SMs through in vitro adventitious root of P. veris other than natural resources. The protocol described would be highly informative and interesting for researchers who work in the production of pharmaceutically important SMs from plants and helps to further standardize to increase the amounts. A few minor corrections are recommended before considering this manuscript for publication.

Minor comments

Line 82: It is not clear how the authors arrived at the concentrations and selection of hormones from the stock to subsequent in vitro root cultures (5.4 μM α-naphthaleneacetic acid (NAA), 0.5 μM kinetin)? Is it adopted from published literature like stock culture (16) or standardized by the authors earlier? Please mention. - Stock cultures including initial establishment, in vitro seed germination and creation of seedlings were established according to the protocol from previous published work on the same plant species-subspecies by Grigoriadou et al. (2020). The concentrations and selection of plant growth regulators types including the amount of carbon source from the shoot-tip explants in agar-solidified medium (shoot proliferation, rooting) to subsequent in vitro root cultures in liquid medium for species-specific secondary metabolites production were earlier standardized by the authors of this study through preliminary trials in other plant species. We did preliminary experiments based up to a degree on references referred to other species with different types and concentrations of plant growth regulators (PGRs) and the response of explants (for example in vitro adventitious roots) was quite better in 5.4 μM NAA + 0.5 μM kinetin combination, thus we decided to use this combined phytohormones treatment in this study with Primula veris subsp. veris as well. The same preliminary examination testing with different PGR types and concentration was followed in the case of shoot-tip cultures in agar-solidified medium for proliferation and rooting prior the dissection of roots from rooted microshoots and the initiation of liquid in vitro root cultures.

Line 86: laminar air blow hood change to laminar air flow hood – Corrected

Line 90: 24h dark, T 22 ± 1°C temperature. What is T represented here? - “temperature” corrected

Line 129: microshoots in vitro and gradually acclimatized... Change to microshoots in vitro was gradually acclimatized - Corrected

Line 190: primeverin YI (0.09-0.10 mg). Check the values mentioned in Table 2 - Rechecked and written accordingly in text based on mean values in the respective Table

Line 191-192. parameters were almost 2- fold higher under 15 oC and 22 oC compared to 10 oC. It is not really almost 2-fold increase. Better check and change the statement accordingly. The whole statement was rechecked and changed accordingly in text based on mean values in the respective Table.

Line 206: P. veris change to Italics – Corrected

Line 214-216: It is interesting to see the soil-grown roots accumulated almost 3-fold higher content of primulic acid. Could you monitor the soil temperature or other parameters like pH that lead to this difference? The aim of the analysis of the soil-grown roots was to detect if the same species-specific secondary metabolites exist and in which amount approximately as an indicator for in vitro produced. The pH was 6 and samples were taken at the middle of October. These days mean soil temperature was approximately 19oC as it has been measured by the nearby Meteorological Station Observatory of Aristotle University of Thessaloniki.

Reviewer 4 Report

The paper titled : “Species-specific secondary metabolites’ production in vitro  from Primula veris subsp. veris adventitious root cultures”, submitted by the authors Sarropoulou et al., investigates  an in vitro adventitious root production protocol in liquid media for the production of four species-specific secondary metabolites.

 The paper contains good amount of data about the selected topic and is if special interest for researchers within this field. There are some things need to be addressed before the publishing of this paper:

1.       In the abstract

-           Please add the amount of major secondary metabolites found in the dry matter using tissue culture technique.

-          Please remove the last paragraph in the abstract to be the conclusion part of the paper.  

-          Add the prospect of the work here and in the conclusion section as well

2.       In the introduction part :

-          Line 55-56 add reference.

-          The novelty of the work need to be highlighted in the introduction and results and discussion parts.

3.       In the materials and methods part

-          Give details of the methods in lines 80-81.

-          Add reference to lines 85-92.

-          Add reference to lines 140- 147.

4.       In the results section,

-          P. veris in line 206 need to be italic , please note that in all the manuscript.

I see from tables 2 and 3, that the tissue culture technique results in relatively lower amount of secondary metabolits, may be improvements should be included here in this study, this should be highlighted in the abstract and conclusion as cons of the this method.

Its much better to merge tables 2 and 3 for comparison reason.

5.       The conclusion is missing.

I give you major revision.

Author Response

Dear Reviewer #4,

We are grateful to receive reviews that help us improve the quality of our manuscript. Following the reviewer #4 comments we corrected the text adopting most of the comments suggested. Please find below our comments to the review. In the revised manuscript, all changes and new additions in text and throughout the manuscript are highlighted with letters of blue color. In addition, in this response letter to your comments, the authors gave responses below each comment in blue color lettering format.

Reviewer #4

Comments and Suggestions for Authors

The paper titled: “Species-specific secondary metabolites’ production in vitro from Primula veris subsp. veris adventitious root cultures”, submitted by the authors Sarropoulou et al., investigates an in vitro adventitious root production protocol in liquid media for the production of four species-specific secondary metabolites. The paper contains good amount of data about the selected topic and is if special interest for researchers within this field. There are some things need to be addressed before the publishing of this paper:

  1. In the abstract

- Please add the amount of major secondary metabolites found in the dry matter using tissue culture technique. Done

-  Please remove the last paragraph in the abstract to be the conclusion part of the paper.  Done

- Add the prospect of the work here and in the conclusion section as well. This study set the basis for further evaluation of elicitors used that could promote further the development of root biomass in vitro and in parallel the mass production biosynthesis of these species-specific secondary metabolites in large scale in bioreactors.

  1. In the introduction part:

-  Line 55-56 add reference.  The missing reference was added

- The novelty of the work need to be highlighted in the introduction and results and discussion parts. Corrected and highlighted

  1. In the materials and methods part

- Give details of the methods in lines 80-81. In the revised manuscript, a more detailed description of methodology applied based on previous published work by Grigoriadou et al. (2020) was provided. “Stock in vitro cultures, derived from germination of 8-year old cold-storage (4oC, RH<5%) seeds, were established following the protocol described by Grigoriadou et al. [16], wherein a 50% germination rate obtained after 60 days of culture in MS [17] medium fortified with 250 mg/L gibberellic acid (GA3), 20 g/L sucrose and 6 g/L Plant Agar (pH:5.8) [16]”

-  Add reference to lines 85-92. “The concentrations and selection of plant growth regulators types including the amount of carbon source from the shoot-tip explants in agar-solidified medium (0.88 μM BA, 0.1 μM IBA, 30 g/L sucrose) to subsequent in vitro root cultures in liquid medium (5.4 μM NAA, 0.5 μM kinetin, 30 g/L sucrose) were earlier standardized by the authors of this study through preliminary trials and experiments, thus they were not adopted from published literature such as stock culture including disinfection, initial establishment and in vitro seed germination and creation of seedlings [16] (Grigoriadou et al. 2020).”          

- Add reference to lines 140-147. The missing reference was added in text and reference list.

  1. In the results section,

-          P. veris in line 206 need to be italic, please note that in all the manuscript. Corrected in this point and throughout the manuscript.

-     I see from tables 2 and 3, that the tissue culture technique results in relatively lower amount of secondary metabolites, may be improvements should be included here in this study, this should be highlighted in the abstract and conclusion as cons of the this method. It has been included both in abstract and in conclusion section

  • Its much better to merge tables 2 and 3 for comparison reason. The aim of the analysis of the soil-grown roots was to detect if the same species-specific secondary metabolites exists and in which amount approximately (as an indicator for in vitro produced compounds). Tables 2 and 3 cannot be merged as statistical analysis cannot be combined, because all grown parameters between in vitro roots and soil grown roots are different (in vitro-soil, duration of culture, different temperatures, root culture – whole plant etc).

  1. The conclusion is missing. A separate Conclusion section was provided highlighting the main advancements achieved and pointing out future prospects of this study.

Round 2

Reviewer 1 Report

Dear author(s),

I strongly recommend you to read my previous comments once again and to address all my notes.

Author Response

Reviewer #1

Comments and Suggestions for Authors

Dear author(s), I strongly recommend you to read my previous comments once again and to address all my notes.

Authors’ response: Dear Reviewer #1,

All the authors are grateful to receive reviews that help us improve the quality of our manuscript. Following your previous comments as well as the specific notes of the Academic Editor we corrected the text adopting most of the comments suggested. In the revised manuscript, all changes and new additions in text and throughout the manuscript are highlighted with green color.

Reviewer 2 Report

All comments were addressed and the manuscript was highly improved.

Author Response

All the authors appreciate the effort paid and the reviewer's comments for improving the scientific background and final image of the manuscript.

Reviewer 4 Report

Accepted for me 

Author Response

(The authors gave the same response as above.)
